# Carob Extract Supplementation Together with Caloric Restriction and Aerobic Training Accelerates the Recovery of Cardiometabolic Health in Mice with Metabolic Syndrome

**DOI:** 10.3390/antiox11091803

**Published:** 2022-09-13

**Authors:** Maria de la Fuente-Fernández, Mario de la Fuente-Muñoz, Marta Román-Carmena, Sara Amor, Ana Belén García-Redondo, Javier Blanco-Rivero, Daniel González-Hedström, Alberto E. Espinel, Ángel Luís García-Villalón, Miriam Granado

**Affiliations:** 1Departamento de Fisiología, Facultad de Medicina, Universidad Autónoma de Madrid, 28029 Madrid, Spain; 2Instituto de Investigación Sanitaria La Paz (IdiPaz), 28029 Madrid, Spain; 3CIBER Enfermedades Cardiovasculares, Instituto de Salud Carlos III, 28029 Madrid, Spain; 4R&D Department, Pharmactive Biotech Products S.L.U., Parque Científico de Madrid, Calle Faraday 7, 28049 Madrid, Spain; 5CIBER Fisiopatología de la Obesidad y Nutrición. Instituto de Salud Carlos III, 28029 Madrid, Spain

**Keywords:** obesity, metabolic syndrome, hypertension, insulin resistance, carob extract, aerobic training, inflammation, antioxidant

## Abstract

Carob, the fruit of *Ceratonia siliqua* L. exerts antidiabetic, anti-inflammatory, and antioxidant effects and could be a useful strategy for the treatment and/or prevention of metabolic syndrome (MetS). The aim of this study was to analyze whether supplementation with a carob fruit extract (CSAT+^®^), alone or in combination with aerobic training, accelerates the recovery of cardiometabolic health in mice with MetS subjected to a caloric restriction. For this purpose, mice were fed with a high fat (58% kcal from fat)/high sugar diet for 23 weeks to induce MetS. During the next two weeks, mice with MetS were switched to a diet with a lower caloric content (25% kcal from fat) supplemented or not with CSAT+^®^ (4.8%) and/or subjected to aerobic training. Both caloric reduction and aerobic training improved the lipid profile and attenuated MetS-induced insulin resistance measured as HOMA-IR. However, only supplementation with CSAT+^®^ enhanced body weight loss, increased the circulating levels of adiponectin, and lowered the plasma levels of IL-6. Moreover, CSAT+^®^ supplementation was the only effective strategy to reduce the weight of epidydimal adipose tissue and to improve insulin sensitivity in the liver and in skeletal muscle. Although all interventions improved endothelial function in aorta segments, only supplementation with CSAT+^®^ reduced obesity-induced hypertension, prevented endothelial dysfunction in mesenteric arteries, and decreased the vascular response of aorta segments to the vasoconstrictor AngII. The beneficial cardiometabolic effects of CSAT+^®^ supplementation, alone or in combination with aerobic training, were associated with decreased mRNA levels of pro-inflammatory markers such as MCP-1, TNFα, IL-1β, and IL-6 and with increased gene expression of antioxidant enzymes, such as GSR, GPX-3, and SOD-1 in the liver, gastrocnemius, retroperitoneal adipose tissue, and aorta. In conclusion, supplementation with CSAT+^®^, alone or in combination with aerobic training, to mice with MetS subjected to caloric restriction for two weeks enhances body weight loss, improves the lipid profile and insulin sensitivity, and exerts antihypertensive effects through its anti-inflammatory and antioxidant properties.

## 1. Introduction

Metabolic syndrome (MetS) is a multifactorial disease characterized by several risk factors, such as dyslipidaemia, abdominal obesity, insulin resistance, and hypertension, that predispose for the development of metabolic and cardiovascular diseases, such as type II diabetes and coronary heart disease, among others [1].

The current lifestyle characterized by the consumption of diets rich in fats and sugars, and the lack of regular physical exercise has caused a dramatic increase in the incidence of this syndrome, both in developed and developing countries, resulting in a huge sanitary and economic burden worldwide [2,3]. 

Although the pathophysiology of MetS is complex and not completely understood, it is known is that insulin resistance is at the center of most of the cardiometabolic alterations associated with this syndrome. Indeed, insulin resistance is involved in several metabolic alterations, e.g., increased hepatic production of very low-density lipoprotein cholesterol (VLDL-c) and serum triglycerides and reduced production and serum concentrations of high-density lipoprotein cholesterol (HDL-c) [4]. Moreover, insulin resistance also plays a pivotal role in the development of cardiovascular alterations, such as endothelial dysfunction, increased proliferation of vascular smooth muscle cells, renal sodium retention, altered membrane ion transport, or sympathetic nervous system hyperactivity [5]. Thus, strategies to fight against MetS necessarily involve reducing the phenomenon of insulin resistance. 

One major contributor for the development of insulin resistance in metabolic syndrome is inflammation. The inflammatory state is highly related to obesity, and specifically to visceral adiposity which is associated with infiltration of immune cells within the adipose tissue resulting in an increased production of pro-inflammatory cytokines and a chronic state of low grade inflammation. [6]. In addition, in this condition, adipocytes decrease their ability to manage lipid storage, resulting in increased lipolysis [7] and hyperlipidemia [6,8]. Both hyperlipidaemia and the chronic exposure to inflammatory cytokines play a pivotal role in the development of insulin resistance in several organs and tissues, such as the skeletal muscle and the liver, by reducing the activation of the phosphoinositide 3-kinase (PI3K)/Akt pathway and producing decreased glucose uptake and hyperglycemia [8]. Moreover, the increased local production of proinflammatory factors also impairs insulin sensitivity in several organs and tissues [9].

Another important factor that contributes to the development of insulin resistance in MetS is the increased production of reactive oxygen species (ROS) due to an increased activity of prooxidant enzymes such as NADPH oxidases (NOX) and/or decreased activity of antioxidant enzymes, such as superoxide dismutase (SOD), glutathione reductase (GSR), or glutathione peroxidase (GPX) [10]. Indeed, treatment with different antioxidants, such as ascorbic acid [11] or flavonoids, has been proven to be effective to counteract, at least in part, the development of insulin resistance in diabetic individuals [12]. 

Among antioxidants is carob, the fruit of the tree *Ceratonia siliqua* L. that contains several bioactive compounds with antioxidant capacity, such as galactomannans, gallic acid, and gallotannins [13]. These compounds are present in the pulp [14] and the pods [15], although the seeds also contain considerable amounts of these products [16,17]. 

In a previous study, our group reported that supplementation with a carob fruit extract, branded under the name CSAT+^®^ (Pharmactive Biotech Products, S.L.U., Madrid, Spain), exerts beneficial effects in mice with metabolic syndrome [18]. These beneficial effects are attributed, at least in part, to the presence of phenolic compounds with potent antioxidant capacity. Moreover, supplementation with CSAT+^®^ reduced the infiltration of macrophages in skeletal muscle and adipose tissue as well as the expression of several proinflammatory markers, demonstrating that this extract also exerts anti-inflammatory effects. In the first study, CSAT+^®^ was found to be a good strategy to prevent the development of cardiometabolic alterations when administered during the fattening period to mice developing MetS. However, the possible use of CSAT+^®^ as a therapeutic agent, once MetS is already established, has not been studied yet. Thus, the aim of the present study was to analyze the possible beneficial effects of CSAT+^®^ in the recovery of the cardiometabolic health after the consumption of a diet rich in fats and sugars. Particularly we aimed to analyze if supplementation with CSAT+^®^, alone or in combination with aerobic training, offers an additional benefit in weight reduction and the improvement of metabolic and cardiovascular function in obese mice subjected to a moderate reduction in caloric intake.

## 2. Material and Methods

### 2.1. Carob Extract (CSAT+^®^)

Samples of proprietary blend of carob (*Ceratonia siliqua* L.) pods and seeds extracts together with fructooligosaccharides (FOS) marketed under the brand CSAT+^®^ were provided by Pharmactive Biotech Products S.L.U. (Madrid, Spain). Samples were standardized to 36–40% of galactomannan fiber, 2–4% of FOS and ≥1% total polyphenols by UV and stored in darkness until addition into the animal chow. 

The chemical characterization of CSAT+^®^, including the analysis of phenolic compounds as well as its total antioxidant capacity, was reported in a previous study [18]. 

### 2.2. Animals

Sixty 16-week-old C57BL/6J mice were housed two per cage and maintained in climate-controlled quarters with a 12 h light cycle and under controlled conditions of humidity (50–60%) and temperature (22–24 °C). The sample size calculation was carried out based on previous studies from our group in which the same parameters were analyzed. The G*Power program was used to perform an a priori analysis of one-way ANOVA with 6 experimental groups. The value assumed for the size effect was 0.65, for the significance (α) was 0.5, and for the power (β) 0.95. All the experiments were conducted according to the European Union Legislation and with the approval of the Animal Care and Ethical Committee of the Community of Madrid (Spain) (PROEX 205.0_20).

As previously described [18], to induce MetS, mice were fed ad libitum either with a standard chow (Chow) or with a high fat/high sucrose (HFHS) diet containing 58% kcal from fat with sucrose (Obese). A weekly control of body weight and solid and liquid intake was performed. After 23 weeks, one group of obese mice remained with the HFHS diet for the next two weeks and the others were subjected to a diet change (DC) in order to reduce their caloric intake. For that purpose, they were fed with a diet containing 25% kcal from fat (Obese-DC) supplemented or not with 4.8% CSAT+^®^ (Obese-DCE), aerobic training (Obese-DCT) or CSAT+^®^ and aerobic training (Obese-DCTE). Aerobic training consisted of running for an hour on a treadmill (9 m/min), 5 days a week over a two-week period. To habituate the animals, training speed and duration were progressively increased during the first week of training: Day 1: 5 m/min for 10 min; Day 2: 9 m/min for 20 min; Day 3: 9 m/min for 35 min; Day 4: 9 m/min for 35 min; Day 5: 9 m/min for 45 min; Day 6 on: 9 m/min for 60 min.

MetS was confirmed by the presence of visceral adiposity, hypertriglyceridemia, hyperglycemia, hypertension, and increased circulating levels of low-density lipoprotein cholesterol (LDL-c).

After the two-week treatment, all animals were injected an overdose of sodium pentobarbital (100 mg/kg) and killed by decapitation after overnight fasting.

### 2.3. Quantification of Triglyceride Content in Hepatic Tissue

Briefly, 100 mg of hepatic tissue were homogenized in 300 µL of phosphate-buffer saline (PBS) and centrifuged at 4 °C for 10 min. The supernatant was used to measure triglycerides with a commercial kit (1001313, Spin React S.A.U, Gerona, Spain) following the manufacturer’s instructions.

To determine the accumulation of lipids in the liver, a staining with Sudan III was performed as previously described [19]. Briefly, liver samples were fixed with 4% paraformaldehyde (PFA) in PBS for 24 h, washed with PBS + 30% sucrose for at least 8 h, and included into Optimal Cutting Temperature compound (OCT). Samples were cut in 3 µm sections and incubated for 24 h with Sudan III. Nuclei were counterstained with hematoxylin for 1 min. Images were acquired using a Leica light microscope fitted with 40 × 0.65 NA objective.

### 2.4. Protein Quantification by Western Blot

Protein content was determined by Western Blot as previously described [20]. Briefly, 100 mg of liver, epididymal adipose tissue and gastrocnemius muscle of each animal were homogenized using RIPA buffer. After centrifugation for 20 min at 12,500 rpm and 4 °C, the supernatant was collected and total protein content was analyzed through Bradford method (Sigma-Aldrich; St. Louis, MO, USA) [21]. For the electrophoresis, 20 μg of protein were loaded in each well of a resolving gel with SDS acrylamide (10%) (Bio-Rad; Hercules, CA, USA). Proteins were transferred to polyvinylidene difluoride (PVDF) membranes (Bio-Rad; Hercules, CA, USA), and transfer efficiency was assessed by Ponceau red dyeing (Sigma-Aldrich; St. Louis, MO, USA). Membranes were then blocked with Tris-buffered saline (TBS) containing 5% (*w/v*) of non-fat dried milk for non-phosphorylated proteins or with 5% (*w/v*) bovine serum albumin (BSA) for phosphorylated proteins and incubated overnight with the appropriate primary antibody (Akt 1:1000; Merck Millipore; Dramstadt, Germany and p-Akt (Ser473) 1:500; Cell Signaling Technology; Danvers, MA, USA). Then, membranes were subsequently washed and incubated with the secondary antibody conjugated with peroxidase (1:2000; Pierce; Rockford, IL, USA) for 90 min. The peroxidase activity was measured by chemiluminescence using BioRad Molecular Imager ChemiDoc XRS System (Hércules, CA, USA). All data are referred to % values from control group (Chow) on each gel.

### 2.5. Adipocyte Size

Adipocyte size was measured as previously described [22]. Briefly, adipose tissue samples were fixed in 4% PFA overnight. Samples were then washed in PBS, dehydrated, and embedded in paraffin wax. Sections (5 μm) were cut on a microtome and mounted into slides. Slides were deparaffinized in xylol, rehydrated and washed in distilled water. Then, nuclei were stained with Harri’s Hematoxylin for a minute. After washing, a counterstained with eosin was performed for 4 min. After dehydrating the samples, images were acquired using a Leica light microscope with a 10x objective (Wetzlar, Germany). Adipocyte size was assessed by measuring the area of adipocyte using FIJI for Windows 36bit (NIH, Bethesda, MA, USA).

### 2.6. RNA Preparation and Quantitative Real-Time PCR

Gene expression was assessed in liver, epididymal adipose tissue, gastrocnemius muscle, and aortic tissue as previously described [23]. Briefly, total RNA was extracted using the Tri-Reagent protocol [24]. cDNA was then synthesized from 1 µg of total RNA using a high-capacity cDNA RT kit (Applied Biosystems, Foster City, CA, USA). 

The mRNA concentrations of interleukin-6 (IL-6) (Mm00446190_m1), interleukin-1 beta (IL-1β) (Mm00434228_m1), tumor necrosis factor-alpha (TNF-α) (Mm00443258_m1), monocyte chemoattractant protein (MCP-1) (Mm00441242_m1), glutathione reductase (GSR) (Mm00439154_m1), NADPH oxidase-4 (NOX-4) (Mm00479246_m1), NADPH oxidase-1 (NOX-1) (Mm00549170_m1), superoxide dismutase 1 (SOD-1) (Mm01344233_g1), Glutathione Peroxidase 3 (GPX3) (Mm00492427_m1), peroxisome proliferator-activated receptor gamma coactivator 1-alpha (PGC-1α) (Mm00447181_m1), and endothelial nitric oxide synthase (eNOS) (Mm00435217_m1) were assessed by quantitative real-time PCR using assay on-demand kits for each gene (Applied Biosystems, Foster City, CA, USA). TaqMan Universal PCR Master Mix (Applied Biosystems, Foster City, CA, USA) was used for amplification according to the manufacturer’s protocol using a Step One System (Applied Biosystems, Foster City, CA, USA). Values were normalized to the housekeeping gene Hypoxanthine Phosphoribosyltransferase 1 (HPRT1) (Mm03024075_m1). To determine the relative expression levels, the ΔΔCT method was used [25]. All data are expressed as % control group (Chow).

### 2.7. Mean Arterial Pressure (MAP) Measurement by the Tail-Cuff System

Mean arterial blood pressure (MBP) measurements were performed in each mouse three times/week the four weeks before sacrifice by tail-cuff plethysmography using a Niprem 645 blood pressure system (Cibertec, Madrid, Spain) as previously described [18]. Briefly, mice were placed in a quiet area (22 ± 2 °C) and habituated to the experimental conditions for at least 7 days. Before measurements, mice were prewarmed to 34 °C for 10–15 min. Then, an occlusion cuff and a sensor were placed at the base of the tail. Next, the occlusion cuff was inflated to 250 mm Hg and deflated over 20 s. Five to six measurements were recorded in each mouse. An average value was calculated for each animal, each day of measurement. The final value was the average among all the measurements performed in each mouse.

### 2.8. Experiments of Vascular Reactivity

For the vascular reactivity experiments, the aorta and superior mesenteric artery were carefully dissected, cut in 2 mm segments, and kept in cold isotonic saline solution. Mesenteric and thoracic aortic segments were used for the vasodilation dose-response curves and abdominal aortic segments for the vasoconstriction dose-response curves. As previously described [26], arterial segments were set up in 4-mL organ bath and changes in isometric force were recorded using a PowerLab data acquisition system (ADInstruments, Colorado Springs, CO, USA). After equilibration for 60–90 min, an optimal passive tension of 1 g was applied. Afterwards, the ability of smooth muscle cells to contract was assessed by adding potassium chloride solution (100 mM KCl, 7447-40-7 Merck Millipore, Burlington, MA, USA). Segments which failed to contract at least 0.5 g in response to KCl were discarded. 

For the vasodilation experiments, the segments were previously contracted with U46619 10^−7.5^ M (Sigma-Aldrich, St. Louis, MO, USA). After the acquisition of a stable tone, dose-response curves in response to sodium nitroprusside (NTP; 10^−9^–10^−5^ M) and acetylcholine (ACh; 10^−9^–10^−4^ M) (Sigma-Aldrich, St. Louis, MO, USA) were performed. To study the effect of oxidative stress in the endothelial function, some segments were preincubated for 30 min with antioxidants (Tempol 10^−4^ M and catalase 10,000 U/mL) before the ACh dose-response curve. Vasodilation in response to insulin was assessed by adding a single dose of insulin (10^−6^ M) to precontracted segments. Relaxation was calculated as % of maximum relaxation (Emax) in response to NTP 10^−5^ M. 

For the vasoconstriction studies, dose–response curves were established in response to norepinephrine (10^−9^–10^−4^ M) and angiotensin-II (10^−11^–10^−6^ M) (Sigma-Aldrich, St. Louis, MO, USA). A contraction in response of each dose of the vasoconstrictors was represented as % contraction to KCl.

### 2.9. Vascular O_2_^*−^ Content

The vascular content of superoxide anion was assessed, as previously described [27], using the oxidative fluorescent dye dihydroethidium (DHE, Invitrogen Life Technologies, Carlsbad, CA, USA; Cat-No. D23107). Briefly, arterial sections were equilibrated for 30 min at 37 °C in Krebs-HEPES buffer. Fresh buffer containing DHE (2 µmol/L) was added to each tissue section, cover-slipped, and incubated in a dark and humidified chamber at 37 °C for 30 min. Images were obtained using a laser scanning confocal microscope (Leica TCS SP5 equipped with a krypton/argon laser, x40 objective; Leica Microsystems, Wetzlar, Germany). Fluorescence was detected with a 568 nm long-pass filter. All samples were processed in the same day and the same imaging setting was used for all experimental conditions. For quantification, mean fluorescence density in the target region was analyzed in 2–3 rings per animal from each experimental group. Data were expressed as % of signal in control arteries.

### 2.10. Statistical Analysis

All values are expressed as means ± standard error of the mean (SEM). One-way ANOVA followed by Bonferroni post-hoc test was used for the statistical data analysis using GraphPad Prism 8.0 (San Diego, CA, USA). A *p*-value of ≤0.05 was considered statistically significant.

## 3. Results

### 3.1. Body Weight Gain and Organ Weights

As expected, the mice consuming the HFHS diet gained more weight than control animals over the 25 weeks of diet consumption (Figure 1A; *p* < 0.001). During the last two weeks, all obese mice subjected to reduced caloric intake lost weight (Figure 1B; *p* < 0.001), with this body weight loss being significantly higher in mice supplemented with CSAT+^®^, alone or in combination with aerobic training (*p* < 0.05 for both). However, aerobic training alone did not accentuate the body weight loss induced by the caloric intake reduction. 

Data of organ weights with respect to the length of the left tibia of mice from the different experimental groups are represented in Table 1.

The change to a diet with a lower caloric content (DC) induced a significant reduction in the weights of liver (*p <* 0.001) and adrenal glands (*p <* 0.01), as well as of some adipose tissue (AT) depots, including the lumbar subcutaneous (*p <* 0.05), the visceral retroperitoneal (*p <* 0.001) and the brown interscapular (*p <* 0.01). However, DC alone did not reduce the weights of visceral epidydimal and aortic perivascular Ats. DC together with CSAT+^®^ supplementation or exercise training, in addition to the above-mentioned parameters, also reduced the weights of perivascular adipose tissue (*p <* 0.01 for both) and spleen (*p <* 0.05 for both). In addition, the reduction of lumbar subcutaneous AT and visceral retroperitoneal AT was more pronounced in these mice than in those only subjected to the DC (*p <* 0.05). However, only CSAT+^®^ supplementation reduced the weight of epidydimal AT (*p <* 0.001) and further reduced the liver weight (*p <* 0.05). Finally, the combination of DC, training, and CSAT+^®^ supplementation produced the same effects on organ weights that CSAT+^®^ supplementation alone, except for the reduction of the weights of epidydimal AT and adrenal glands and induced a significant decrease in the weights of soleus and gastrocnemius muscles compared to both obese mice (*p <* 0.01) and obese mice subjected to DC (*p <* 0.05).

### 3.2. Food and Caloric Intake

Data of food intake and caloric intake over the treatment period (weeks 23–25) are represented in Figure 1C,D, respectively.

Compared to controls, food intake was significantly reduced in mice consuming the hypercaloric diets (*p <* 0.05), and this food intake was significantly lower in mice supplemented with CSAT+^®^, alone (*p <* 0.01) or in combination with aerobic training (*p <* 0.05).

Despite reduced food intake, the caloric intake was significantly higher in Obese mice compared to controls (*p* < 0.05) whereas DC and DCT obese mice showed similar caloric intake than control animals. Obese-DCE and Obese-DCTE mice showed reduced caloric intake compared to Obese mice (*p* < 0.05 for both).

### 3.3. Serum Parameters, Adiposity Index and HOMA-IR

As shown in Table 2, MetS was associated with a significant increase in the circulating levels of total lipids, triglycerides, total cholesterol, LDL-c, HDL-c, leptin, insulin (*p <* 0.001 for all), glucose (*p <* 0.05) and IL-6 (*p <* 0.01) and with a significant reduction in the serum levels of adiponectin (*p <* 0.001). Moreover, obese mice showed an increased adiposity index and Homeostatic Model Assessment for Insulin Resistance index (HOMA-IR) compared to mice fed with chow (*p <* 0.001 for both). All the treatments were effective attenuating the obesity-induced alterations in the adiposity index and HOMA-IR as well as in most of the serum parameters. However, only obese mice subjected to DC and supplemented with CSAT+^®^ (DCE) showed decreased circulating levels of IL-6 and increased serum concentrations of adiponectin (*p <* 0.05). Moreover, obese mice subjected to DC and supplemented with CSAT+^®^, alone or in combination with aerobic training, showed decreased concentrations of leptin (*p <* 0.05), total cholesterol (*p <* 0.05 and LDL-c (*p <* 0.01) compared to mice only subjected to DC. 

### 3.4. Mean Arterial Pressure (MAP) 

Obesity was associated with a significant increase in MAP (*p <* 0.001; Table 2). DC alone or combined with aerobic training did not reduce MAP whereas the combination of DC with CSAT+^®^ supplementation induced a significant reduction in MAP (*p <* 0.001). Moreover, MAP from DCE mice was significantly lower than MAP form DC mice (*p <* 0.05). However, the combination of DC, CSAT+^®^ and aerobic training did not produce significant effects on blood pressure (BP) values. 

### 3.5. Activation of PI3K/Akt Pathway in Liver, Gastrocnemius and Retroperitoneal Adipose Explants in Response to Insulin

The p-Akt/Akt ratio in explants of liver, gastrocnemius muscle, and retroperitoneal adipose tissue incubated in the presence/absence of insulin is shown in Figure 2A–C, respectively.

In the absence of insulin, no differences were found in the p-Akt/Akt ratio among experimental groups in none of the tissues. Incubation with insulin significantly increased the p-Akt/Akt ratio in liver, gastrocnemius muscle and AT explants from lean mice (*p <* 0.05 for all). In addition, insulin also increased the p-Akt/Akt ratio in liver explants from obese mice subjected to DC and supplemented with CSAT+^®^, alone or together with aerobic training (*p <* 0.05 for both). However, in gastrocnemius muscle, the p-Akt/Akt ratio was only increased in explants from obese mice supplemented with CSAT+^®^ alone (*p <* 0.01).

### 3.6. Lipid Content and Gene Expression of Pro-Inflammatory and Oxidative Stress Related Markers in Hepatic Tissue

Obesity was associated with a significant increase in the content of triglycerides in hepatic tissue (Figure 3; *p <* 0.001) that was prevented by DCE (*p <* 0.05), DCT (*p <* 0.05), and DCET (*p <* 0.01).

The Increase in lipid content in the liver of obese mice was associated with an upregulation of the pro-inflammatory genes MCP-1 (Figure 4; *p <* 0.001), IL-1β (Figure 4; *p <* 0.001), IL-6 (Figure 4; *p <* 0.05), and TNF-α (Figure 4; *p <* 0.05). DC only reduced the mRNA levels of MCP-1 (*p <* 0.01), whereas DCE also decreased the gene expression of IL-1β (*p <* 0.01) and TNF-α (*p <* 0.05). DCT prevented the obesity-induce upregulation of MCP-1 (*p <* 0.001) and IL-6 (*p <* 0.01) and DCTE also reduced the gene expression of TNF-α (*p <* 0.05). Moreover, the hepatic mRNA levels of TNF-α in DCTE mice were significantly lower than in DCE and DCT mice (*p <* 0.05 for both).

Regarding the gene expression of oxidative stress-related markers, the hepatic gene expression of NOX-1 and GPX-3 was upregulated in obese mice (*p <* 0.05 and *p <* 0.01, respectively), whereas the mRNA levels of NOX-4 (*p <* 0.05), GSR (*p <* 0.05) and SOD-1 (*p <* 0.01) were significantly reduced. DC only attenuated the obesity-induced downregulation of NOX-4 *(p <* 0.05)*,* whereas DCE also prevented the alterations in the mRNA levels of NOX-4 (*p <* 0.01), GSR (*p <* 0.05) and GPX-3 (*p <* 0.01). Finally, DCT attenuated the obesity induced-changes in the gene expression of NOX-1 (*p <* 0.05), NOX-4 (*p <* 0.01), and GPX-3 (*p <* 0.05) and DCTE the changes in NOX-1 and NOX-4 (*p <* 0.05 for both). The gene expression of PGC-1α was significantly lower in Obese-DCTE mice compared to controls (*p* < 0.05).

### 3.7. Gene Expression of Pro-Inflammatory and Oxidative Stress Related Markers in Gastrocnemius Muscle

In gastrocnemius muscle, there was an upregulation in the mRNA levels of MCP-1 (Figure 5A; *p <* 0.001), IL-1β (Figure 5B; *p <* 0.01), IL-6 (Figure 5C; *p <* 0.05), and TNF-α (Figure 5D; *p <* 0.01) in obese mice compared to controls that was not prevented with the DC. Both DCE and DCT significantly decreased the obesity-induced overexpression of IL-1β (*p <* 0.01 and *p <* 0.05, respectively). However only supplementation with CSAT+^®^, alone or in combination with aerobic training, reduced the gene expression of MCP-1 (*p <* 0.01 and *p <* 0.001, respectively). 

Obesity was also associated with a downregulation in the gene expression of NOX-4 (Figure 5F; *p <* 0.05) and with a significant increase in the mRNA levels of GSR (Figure 5G; *p <* 0.01), SOD-1 (Figure 5I; *p <* 0.001) and PGC-1α (Figure 5J; *p <* 0.05) in gastrocnemius muscle. None of these alterations were attenuated by DC alone. On the contrary, DCT downregulated the obesity-induced overexpression of GSR (*p <* 0.05) and increased the mRNA levels of GPX-3 (*p <* 0.05). DCE prevented the changes in the gene expression of GSR, PGC-1α and NOX-4 (*p <* 0.05 for all) induced by obesity and also increased the mRNA levels of GPX-3 (*p <* 0.05). Finally, DCTE reduced the obesity-induced alterations in the gene expression of GSR (*p <* 0.01), SOD-1 (*p <* 0.05) and PGC-1α and up-regulated the mRNA levels of GPX-3 (*p <* 0.05).

### 3.8. Adipocyte Area and Gene Expression of Pro-Inflammatory and Oxidative Stress Related Markers in Retroperitoneal Adipose Tissue

All treatments equally prevented the obesity-induced increase in the adipocyte area (Figure 6; *p <* 0.05) and in the mRNA levels of MCP-1 (Figure 7A; *p <* 0.001 for all) and IL-6 (Figure 7C; *p <* 0.01 for all) in retroperitoneal adipose tissue. However only DCTE significantly reduced the gene expression of IL-1β (*p <* 0.05). Moreover, only obese mice subjected to DC and supplemented with CSAT+^®^, either alone or together with training, showed increased expression of GPX-3 and PGC-1α (*p <* 0.05 for both).

### 3.9. Vascular Function

The maximum vascular response to acetylcholine in the presence/absence of the antioxidants Tempol-Catalase is shown in Figure 8A. 

Obesity was associated with a decreased vascular response to ACh (*p <* 0.05) that was prevented after the incubation of aorta segments with the antioxidants (*p <* 0.05). In the rest of experimental groups, no changes were found in the vascular response to ACh, neither in the absence nor in the presence of the antioxidant.

The maximum vascular response to the vasoconstrictors noradrenaline (in presence/absence of N ω-Nitro-L-arginine methyl ester hydrochloride (L-NAME)) and angiotensin II (AngII) are shown in Figure 8B,C, respectively.

In the absence of L-NAME no significant differences were found among experimental groups. However, preincubation with L-NAME induced a significant increase in the vascular response to NA only in aorta segments from control mice (*p <* 0.001) and in aorta segments from obese mice subjected to DCE (*p <* 0.05) and DCTE (*p <* 0.01). Aorta segments from obese mice showed an increased vascular response to Ang II compared to aorta segments from control mice (*p <* 0.05). None of the interventions prevented the increased vasoconstriction in response to Ang II except for DCE (*p <* 0.05). 

Aorta segments from obese mice showed a decreased vascular response to insulin compared to aorta segments from control mice (Figure 8D; *p <* 0.05). None of the interventions prevented the obesity-induced vascular insulin resistance except for DCTE (*p <* 0.05). Finally, the results in segments of mesenteric arteries (Figure 8E) indicate a reduced endothelium-dependent relaxation in segments from obese mice compared to controls that was only prevented by DCE (*p <* 0.05)

### 3.10. Gene Expression of Pro-Inflammatory and Oxidative Stress Related Markers in Aortic Tissue

No significant changes were found in the gene expression of pro-inflammatory markers in aortic tissue except for an increase in the mRNA levels of MCP-1 (Figure 9A; *p <* 0.05) and IL-1β in obese DCT mice (Figure 9A; *p <* 0.05) that was prevented by CSAT+^®^ supplementation (*p <* 0.05).

Obesity induced a significant decrease in the mRNA levels of NOX-4 (Figure 9F; *p <* 0.05), GSR (Figure 9G; *p <* 0.01), GPX-3 (Figure 9H; *p <* 0.05), and SOD-1 (Figure 9I; *p <* 0.05). DC attenuated the reduction in the gene expression of GSR and SOD-1 (*p <* 0.05 for both), whereas DCE, DCT, and DCTE completely prevented the obesity-induced reduction in the mRNA levels of NOX-4, GSR, GPX-3 and SOD-1 (*p <* 0.05). Finally, aorta segments from obese mice showed decreased gene expression of eNOS (Figure 9J; *p <* 0.01), and this effect was prevented by all interventions except for DC (*p <* 0.01).

### 3.11. Vascular Content of Superoxide Anion in Arterial Tissue

The content of superoxide anion was significantly higher in aorta sections from Obese and Obese-DCTE mice compared to Chow mice (Figure 10; *p <* 0.05). However, there were no differences among the content of O_2_^*−^ in the rest of the experimental groups, except for aorta sections from Obese-DCE mice that showed decreased content of O_2_^*−^ compared to Obese mice (*p <* 0.05).

## 4. Discussion

In this paper, we describe for the first time the beneficial effects of supplementation with the carob fruit extract CSAT+^®^ in body weight reduction and the recovery of cardiometabolic health in obese mice subjected to a reduction in the caloric intake. Moreover, the study includes an experimental group of obese mice subjected to both diet change and aerobic training as a positive control and another group of mice subjected to both interventions, to assess if training and CSAT+^®^ supplementation exert synergistic effects on the improvement of metabolic and cardiovascular function.

Our results show that the change from a hypercaloric diet (58% kcal from fat) to a diet with a lower caloric content (25% kcal from fat) for a two-week period, as expected, induced a significant reduction in body weight and that this body weight loss was more accentuated when the obese mice were supplemented with CSAT+^®^, alone or in combination with aerobic training. This result agrees with a previous study in humans in which carob supplementation for six weeks induces a significant reduction in body weight in young taekwondo athletes [28]. However, results from clinical trials with obese individuals show that supplementation with a snack enriched with carob did not reduce neither body weight nor adiposity [29]. Discrepancies among studies could be due not only to differences in dosages and durations of the treatments, but also to differences in the type of diets and the nutritional state of the subjects. Indeed, we reported in a previous study that supplementation with CSAT+^®^ did not have an impact on body weight gain in mice fed with a high fat/high sugar diet for 20 weeks [18], indicating that the positive effect of CSAT+^®^ on body weight reduction possibly depends on the caloric content of the diet.

The beneficial effect of CSAT+^®^ improving body weight loss is most likely due to a satiating effect, since obese mice supplemented with the carob extract showed decreased food intake compared to non-supplemented mice. Likewise, other authors have reported that the consumption of carob-derived products is effective at inducing satiety and decreasing food intake in both lean [30] and in obese individuals [29] by affecting the postprandial levels of glucose, insulin, or ghrelin [30,31,32].

The decrease in body weight after the caloric intake reduction was associated with a significant decrease in the weights of liver and adrenal glands in all groups. However, DC alone was not sufficient to reduce the obesity-induced splenomegaly. DC was effective in reducing adiposity in most adipose tissue depots, but only supplementation with CSAT+^®^ significantly decreased the weight of epidydimal adipose tissue. Moreover, mice supplemented with CSAT+^®^ and performing aerobic training also showed a decrease in the weight of kidneys, pointing to a synergistic effect between both interventions. The anti-obesity effect of carob is already reported [33] and is due, at least in part, to a direct inhibitory effect on adipocyte differentiation [34]. However, supplementation with CSAT+^®^ together with a high fat/high sucrose diet did not have an impact on adiposity [18], suggesting again that the effects of CSAT+^®^ on body fat may be dependent on the caloric content of the diet.

Calorie reduction significantly improved several serum parameters in obese mice such as the lipidic profile, the circulating levels of glucose, leptin and insulin and the HOMA-IR. However, only mice subjected to aerobic training or CSAT+^®^ supplementation showed reduced circulating levels of triglycerides. The positive effects of carob supplementation in improving the lipid profile have been already reported both in obese experimental animals [18,35,36,37] and in humans [29,38,39]. Furthermore, the hypolipidemic effect is not only present in obese individuals, but also in healthy subjects [40,41,42], and is related, at least in part, to the capacity of carob to reduce the binding of lipids to bile acids, thus reducing lipid absorption [43,44]. Similarly, it is reported that some of the compounds identified in CSAT+^®^, such as quercetin and gallic acid [18], also exert beneficial effects on the lipid profile in overweight subjects [45] and in obese rodents [46,47].

Although all mice subjected to diet change showed an improved lipid profile, only obese mice supplemented with CSAT+^®^, either alone or together with aerobic training, showed decreased serum concentrations of IL-6 and increased circulating levels of adiponectin compared to non-treated obese mice. These results agree with those reported in our previous study in which CSAT+^®^ was administered as a preventive strategy to mice during the development of MetS [18]. Furthermore, other studies in which carob was administered as a therapeutic strategy, both in prediabetic obese humans [48] and in obese mice [49], have also found decreased circulating serum levels of IL-6, pointing to a clear anti-inflammatory effect of this product. Based on previous studies, this anti-inflammatory effect may be conferred, at least in part, by certain compounds present in CSAT+^®^ such as some flavonols and gallic acid [46,50].

An important finding of this work is that, even though all treatments reduced the obesity-induced increase in the HOMA-IR, only obese mice supplemented with CSAT+^®^ showed preserved insulin sensitivity in the liver and in skeletal muscle. This fact was also present in our previous study [18], and may be related to the lower circulating levels of IL-6, since the low grade of chronic inflammation is one of the major factors responsible for the development of insulin resistance in a context of obesity [51]. Another factor that is most likely involved in the decreased insulin resistance in skeletal muscle and hepatic tissue is adiponectin, a circulating hormone with insulin-sensitizing effects [52] that, as previously described [18], is increased in obese mice supplemented with CSAT+^®^. The beneficial effects of carob in reducing insulin resistance, and thus improving the glycemic state, have been widely described, not only in animal models [18,53,54,55], but also in humans [48,56,57]. This effect is most likely related to decreased carbohydrate absorption [58,59] and could be mediated by D-pinitol, a bioactive compound present in carob with potent antidiabetic effects [60]. Moreover, since oxidative stress also plays a pivotal role in the development of insulin resistance [61], the presence of antioxidant compounds in CSAT+^®^, such as gallotannins and flavonols [18], may also be involved in its insulin sensitizing effects in peripheral tissues. Indeed, gallic acid and quercetin are reported to improve glucose tolerance both in humans and animal models of diet-induced obesity [46,62,63]. Indeed, our results indicate that, opposite to caloric reduction alone, supplementation with carob extract not only reduces the gene expression of certain pro-inflammatory markers, such as MCP-1 or IL-1β in skeletal muscle and hepatic tissues, but also the obesity-induced alterations in the mRNA levels of oxidative stress-related markers. In addition, for some markers, such as TNF-α and NOX-1 in the liver or SOD-1 in the gastrocnemius, this positive effect is more pronounced in mice supplemented with CSAT+^®^ and subjected to aerobic training, pointing to a synergistic effect between both interventions. These results agree with previous studies reporting that carob supplementation exerts anti-inflammatory and antioxidants effects in hepatic and muscular tissues [18,64]. However, this effect was not found in visceral adipose tissue in which CSAT+^®^ supplementation did not show additional benefits to caloric reduction, except for increased mRNA levels of the antioxidant enzyme GPX-3. Moreover, the triglyceride content in the liver and the adipocyte size in retroperitoneal adipose tissue was also similar in all obese mice subjected to the diet change for two weeks, regardless of supplementation or not with CSAT+^®^ or aerobic training. These results indicate that improved insulin sensitivity in CSAT+^®^ supplemented mice is most likely related to circulating factors, such as IL-6 and adiponectin, as well as to the reduced local expression of inflammatory and oxidative stress related markers, rather than to decreased adiposity or lipidic content. In this regard, in our previous study, we also found a significant reduction of insulin resistance, together with increased circulating levels of adiponectin and decreased concentrations of IL-6 in mice with MetS supplemented with CSAT+^®^, without significant changes in adiposity [18].

In addition to the beneficial effects on body composition, lipid profile, and insulin sensitivity, our results also show that CSAT+^®^ supplementation in obese mice subjected to caloric reduction over a two-week period is the only intervention of all tested in this study capable of reducing arterial pressure. The antihypertensive effect of carob has been described previously [18,37,65] and is related, at least in part, to a decreased vascular response to the vasoconstrictor AngII [18]. Moreover, as previously reported [18,37], our results also show that carob supplementation prevents obesity-induced endothelial dysfunction in the aorta. This effect may be produced, at least in part, by quercetin which exerts vasodilating effects through the increase in NO availability [66,67]. Although this effect is also produced by caloric reduction and aerobic training in aorta segments, only CSAT+^®^ supplementation prevents endothelial dysfunction in mesenteric arteries. Moreover, only supplementation with CSAT+^®^, alone or in combination with exercise, significantly increases the vasoconstriction to NA in the presence of L-NAME. The results obtained after the preincubation of aorta rings with the antioxidants tempol and catalase demonstrates that the beneficial effects of both exercise and CSAT+^®^ supplementation on endothelial function are due to their antioxidant effects. Indeed, both interventions significantly increased the gene expression of antioxidant enzymes in arterial tissue as well as the mRNA levels of eNOS. Similarly, supplementation with CSAT+^®^ prevented the obesity-induced alterations in the gene expression of oxidative stress related markers in arterial tissue of mice with MetS [18].

Finally, our results show that the combination of CSAT+^®^ supplementation and aerobic training is the only effective strategy to prevent obesity-induced vascular insulin resistance, pointing again to a synergistic effect between them. These results disagree with those reported in previous studies in which both training [68] or CSAT+^®^ supplementation alone [18] increase insulin-induced vasodilation. These discrepancies could be due to differences in the species (mice vs. humans), type of exercise (aerobic vs. resistance), or to the duration of the supplementation with CSAT+^®^ (two weeks vs. twenty weeks).

## 5. Conclusions

In conclusion, caloric reduction and supplementation with the carob extract CSAT+^®^, alone or in the presence of aerobic training, is a good strategy for the recovery of cardiometabolic health in mice with metabolic syndrome. Both interventions, CSAT+^®^ supplementation and aerobic training, exert synergistic effects in preventing the obesity-induced increase in kidney weight, vascular insulin resistance, and changes in some pro-inflammatory and oxidative genes in the liver and the gastrocnemius. Nevertheless, CSAT+^®^ supplementation alone is sufficient to reduce most of the metabolic alterations associated to MetS and is the only effective strategy to reduce blood pressure.

## Figures and Tables

**Figure 1 antioxidants-11-01803-f001:**
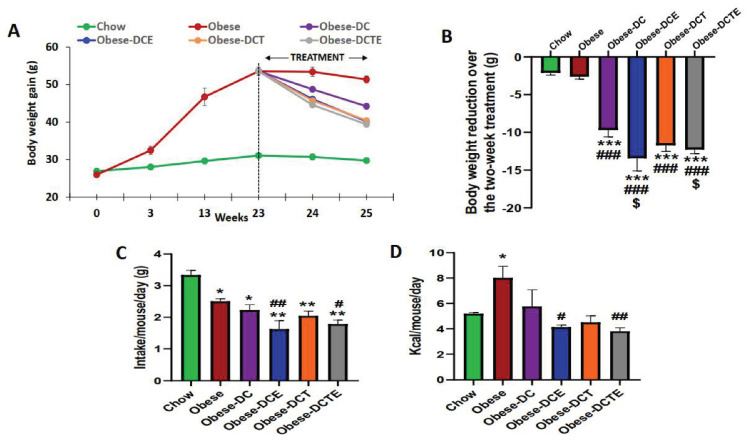
Body weight gain (**A**); Body weight reduction during the two weeks of treatment (week 23 to week 25) (**B**); Food intake during the two weeks of treatment (g/mouse/day); (**C**) Caloric intake during the two weeks of treatment (Kcal/mouse/day) (**D**). Chow = mice fed a standard diet for 25 weeks; Obese = mice fed a high fat (60% kcal from fat)/high sucrose diet for 25 weeks; DC = mice fed a high fat (60% kcal from fat)/high sucrose diet for 23 weeks and switched to a diet with lower caloric intake (25% kcal from fat) for two weeks; DCE = mice fed a high fat (60% kcal from fat)/high sucrose diet for 23 weeks and switched to a diet with lower caloric intake (25% kcal from fat) supplemented with CSAT+^®^ for two weeks; DCT = mice fed a high fat (60% kcal from fat)/high sucrose diet for 23 weeks and switched to a diet with lower caloric intake (25% kcal from fat) and subjected to aerobic training for two weeks; DCTE = mice fed a high fat (60% kcal from fat)/high sucrose diet for 23 weeks and switched to a diet with lower caloric intake (25% kcal from fat) supplemented with CSAT+^®^ and subjected to aerobic training for two weeks. Values are represented as mean ± SEM; *n* = 8–10 mice/group. * *p <* 0.05 vs. chow; ** *p <* 0.01 vs. chow; *** *p <* 0.001 vs. chow; # *p <* 0.05 vs. obese; ## *p <* 0.01 vs. obese; ### *p <* 0.001 vs. obese; $ *p <* 0.05 vs. obese-DC.

**Figure 2 antioxidants-11-01803-f002:**
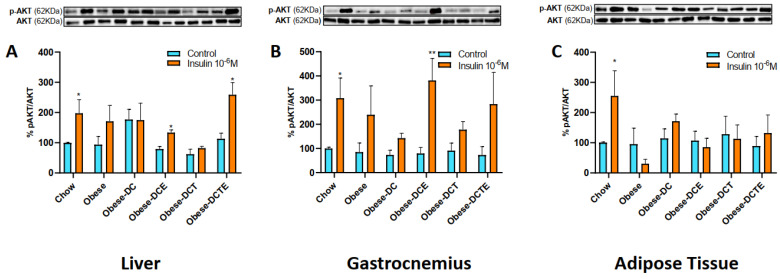
p-Akt/Akt ratio in explants of liver (**A**), gastrocnemius (**B**) and adipose tissue (**C**) incubated in the presence/absence of insulin (10–6M). Chow = mice fed a standard diet for 25 weeks; Obese = mice fed a high fat (60% kcal from fat)/high sucrose diet for 25 weeks; DC = mice fed a high fat (60% kcal from fat)/high sucrose diet for 23 weeks and switched to a diet with lower caloric intake (25% kcal from fat) for two weeks; DCE = mice fed a high fat (60% kcal from fat)/high sucrose diet for 23 weeks and switched to a diet with lower caloric intake (25% kcal from fat) supplemented with CSAT+^®^ for two weeks; DCT = mice fed a high fat (60% kcal from fat)/high sucrose diet for 23 weeks and switched to a diet with lower caloric intake (25% kcal from fat) and subjected to aerobic training for two weeks; DCTE = mice fed a high fat (60% kcal from fat)/high sucrose diet for 23 weeks and switched to a diet with lower caloric intake (25% kcal from fat) supplemented with CSAT+^®^ and subjected to aerobic training for two weeks. Values are represented as mean ± SEM; *n* = 8–10 mice/group. * *p <* 0.05 vs. control; ** *p <* 0.01 vs. control.

**Figure 3 antioxidants-11-01803-f003:**
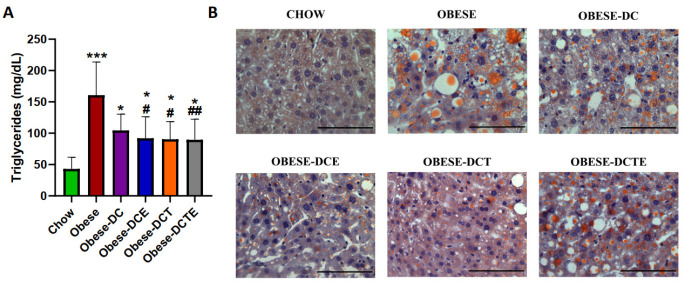
Quantification of triglycerides in the liver (**A**) and representative images of liver sections Scheme 100 M (**B**). Chow = mice fed a standard diet for 25 weeks; Obese = mice fed a high fat (60% kcal from fat)/high sucrose diet for 25 weeks; DC = mice fed a high fat (60% kcal from fat)/high sucrose diet for 23 weeks and switched to a diet with lower caloric intake (25% kcal from fat) for two weeks; DCE = mice fed a high fat (60% kcal from fat)/high sucrose diet for 23 weeks and switched to a diet with lower caloric intake (25% kcal from fat) supplemented with CSAT+^®^ for two weeks; DCT = mice fed a high fat (60% kcal from fat)/high sucrose diet for 23 weeks and switched to a diet with lower caloric intake (25% kcal from fat) and subjected to aerobic training for two weeks; DCTE = mice fed a high fat (60% kcal from fat)/high sucrose diet for 23 weeks and switched to a diet with lower caloric intake (25% kcal from fat) supplemented with CSAT+^®^ and subjected to aerobic training for two weeks. Values are represented as mean ± SEM; *n* = 8–10 mice/group. * *p <* 0.05 vs. chow; *** *p <* 0.001 vs. chow; # *p <* 0.05 vs. obese; ## *p <* 0.01 vs. obese.

**Figure 4 antioxidants-11-01803-f004:**
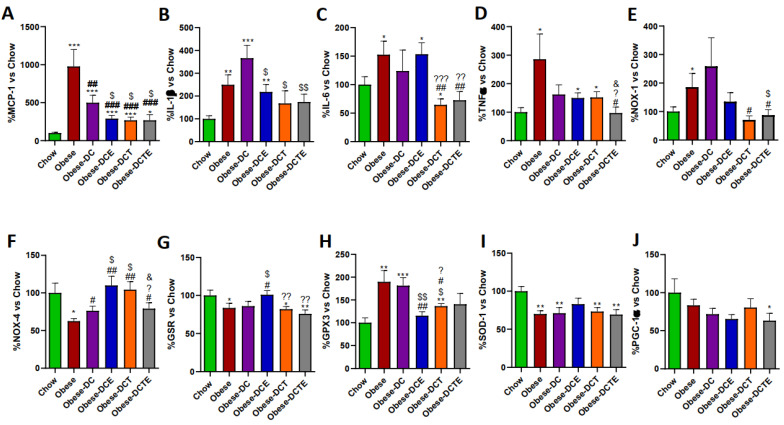
Hepatic gene expression of Monocyte Chemotactic Protein-1 (**A**), Interleukin 1β (**B**), Interleukin 6 (**C**), Tumor Necrosis Factor α (**D**) NADPH oxidase 1 (**E**), NADPH oxidase 4 (**F**), Glutathione Reductase (**G**), Glutathione Peroxidase 3 (**H**), Super Oxide Dismutase 1 (**I**) and PPAR-γ coactivator 1α (**J**). Chow = mice fed a standard diet for 25 weeks; Obese = mice fed a high fat (60% kcal from fat)/high sucrose diet for 25 weeks; DC = mice fed a high fat (60% kcal from fat)/high sucrose diet for 23 weeks and switched to a diet with lower caloric intake (25% kcal from fat) for two weeks; DCE = mice fed a high fat (60% kcal from fat)/high sucrose diet for 23 weeks and switched to a diet with lower caloric intake (25% kcal from fat) supplemented with CSAT+^®^ for two weeks; DCT = mice fed a high fat (60% kcal from fat)/high sucrose diet for 23 weeks and switched to a diet with lower caloric intake (25% kcal from fat) and subjected to aerobic training for two weeks; DCTE = mice fed a high fat (60% kcal from fat)/high sucrose diet for 23 weeks and switched to a diet with lower caloric intake (25% kcal from fat) supplemented with CSAT+^®^ and subjected to aerobic training for two weeks. Values are represented as mean ± SEM; *n* = 8–10 mice/group. * *p <* 0.05 vs. chow; ** *p <* 0.01 vs. chow; *** *p <* 0.001 vs. chow; # *p <* 0.05 vs. obese; ## *p <* 0.01 vs. obese; ### *p <* 0.001 vs. obese; $ *p <* 0.05 vs. obese-DC; $$ *p <* 0.05 vs. obese-DC, ? *p <* 0.05 vs. obese-DCE; ?? *p <* 0.01 vs. obese-DCE; ??? *p <* 0.001 vs. obese-DCE; & *p <* 0.05 vs. obese-DCT.

**Figure 5 antioxidants-11-01803-f005:**
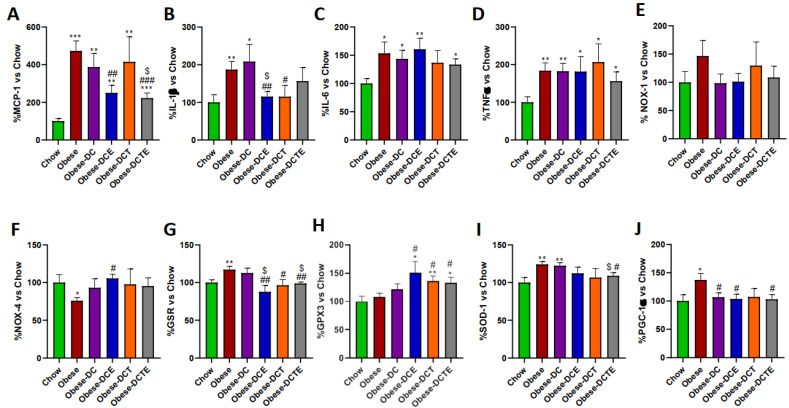
Gene expression of Monocyte Chemotactic Protein-1 (**A**), Interleukin 1β (**B**), Interleukin 6 (**C**), Tumor Necrosis Factor α (**D**) NADPH oxidase 1 (**E**), NADPH oxidase 4 (**F**), Glutathione Reductase (**G**), Glutathione Peroxidase 3 (**H**), Super Oxide Dismutase 1 (**I**) and PPAR-γ coactivator 1α (**J**) in gastrocnemius muscle. Chow = mice fed a standard diet for 25 weeks; Obese = mice fed a high fat (60% kcal from fat)/high sucrose diet for 25 weeks; DC = mice fed a high fat (60% kcal from fat)/high sucrose diet for 23 weeks and switched to a diet with lower caloric intake (25% kcal from fat) for two weeks; DCE = mice fed a high fat (60% kcal from fat)/high sucrose diet for 23 weeks and switched to a diet with lower caloric intake (25% kcal from fat) supplemented with CSAT+^®^ for two weeks; DCT = mice fed a high fat (60% kcal from fat)/high sucrose diet for 23 weeks and switched to a diet with lower caloric intake (25% kcal from fat) and subjected to aerobic training for two weeks; DCTE = mice fed a high fat (60% kcal from fat)/high sucrose diet for 23 weeks and switched to a diet with lower caloric intake (25% kcal from fat) supplemented with CSAT+^®^ and subjected to aerobic training for two weeks. Values are represented as mean ± SEM; *n* = 8–10 mice/group. * *p <* 0.05 vs. chow; ** *p <* 0.01 vs. chow; *** *p <* 0.001 vs. chow; # *p <* 0.05 vs. obese; ## *p <* 0.01 vs. obese; ### *p <* 0.001 vs. obese; $ *p <* 0.05 vs. obese-DC.

**Figure 6 antioxidants-11-01803-f006:**
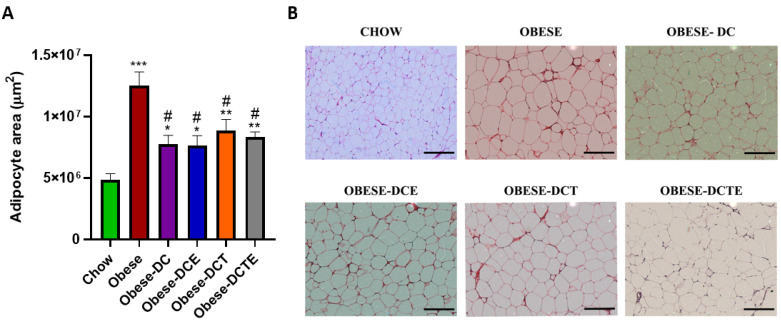
Size of adipocytes from visceral adipose tissue (**A**) and representative images of H/E dying in sections of visceral adipose tissue (**B**). The scale bar is equivalent to 200μm. Chow = mice fed a standard diet for 25 weeks; Obese = mice fed a high fat (60% kcal from fat)/high sucrose diet for 25 weeks; DC = mice fed a high fat (60% kcal from fat)/high sucrose diet for 23 weeks and switched to a diet with lower caloric intake (25% kcal from fat) for two weeks; DCE = mice fed a high fat (60% kcal from fat)/high sucrose diet for 23 weeks and switched to a diet with lower caloric intake (25% kcal from fat) supplemented with CSAT+^®^ for two weeks; DCT = mice fed a high fat (60% kcal from fat)/high sucrose diet for 23 weeks and switched to a diet with lower caloric intake (25% kcal from fat) and subjected to aerobic training for two weeks; DCTE = mice fed a high fat (60% kcal from fat)/high sucrose diet for 23 weeks and switched to a diet with lower caloric intake (25% kcal from fat) supplemented with CSAT+^®^ and subjected to aerobic training for two weeks. Values are represented as mean ± SEM; *n* = 8–10 mice/group. * *p <* 0.05 vs. chow; ** *p <* 0.01 vs. chow; *** *p <* 0.001 vs. chow; # *p <* 0.05 vs. obese.

**Figure 7 antioxidants-11-01803-f007:**
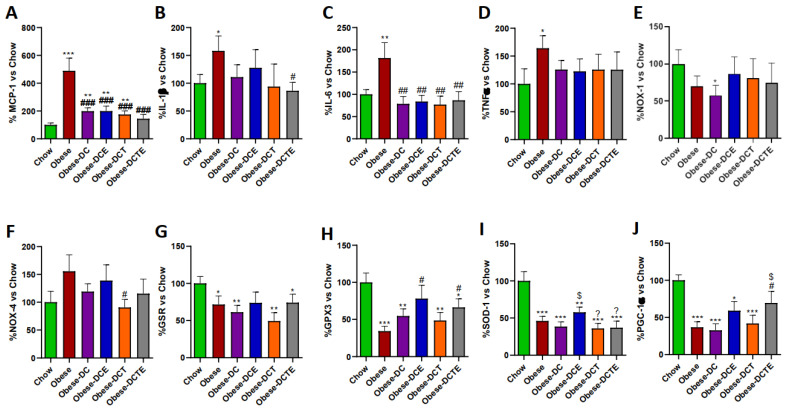
Gene expression of Monocyte Chemotactic Protein-1 (**A**), Interleukin 1β (**B**), Interleukin 6 (**C**), Tumor Necrosis Factor α (**D**) NADPH oxidase 1 (**E**), NADPH oxidase 4 (**F**), Glutathione Reductase (**G**), Glutathione Peroxidase 3 (**H**), Super Oxide Dismutase 1 (**I**) and PPAR-γ coactivator 1α (**J**) in visceral adipose tissue. Chow = mice fed a standard diet for 25 weeks; Obese = mice fed a high fat (60% kcal from fat)/high sucrose diet for 25 weeks; DC = mice fed a high fat (60% kcal from fat)/high sucrose diet for 23 weeks and switched to a diet with lower caloric intake (25% kcal from fat) for two weeks; DCE = mice fed a high fat (60% kcal from fat)/high sucrose diet for 23 weeks and switched to a diet with lower caloric intake (25% kcal from fat) supplemented with CSAT+^®^ for two weeks; DCT = mice fed a high fat (60% kcal from fat)/high sucrose diet for 23 weeks and switched to a diet with lower caloric intake (25% kcal from fat) and subjected to aerobic training for two weeks; DCTE = mice fed a high fat (60% kcal from fat)/high sucrose diet for 23 weeks and switched to a diet with lower caloric intake (25% kcal from fat) supplemented with CSAT+^®^ and subjected to aerobic training for two weeks. Values are represented as mean ± SEM; *n* = 8–10 mice/group. * *p <* 0.05 vs. chow; ** *p <* 0.01 vs. chow; *** *p <* 0.001 vs. chow; # *p <* 0.05 vs. obese; ## *p <* 0.01 vs. obese; ### *p <* 0.001 vs. obese; $ *p <* 0.05 vs. obese-DC. ? *p <* 0.05 vs. obese-DCE.

**Figure 8 antioxidants-11-01803-f008:**
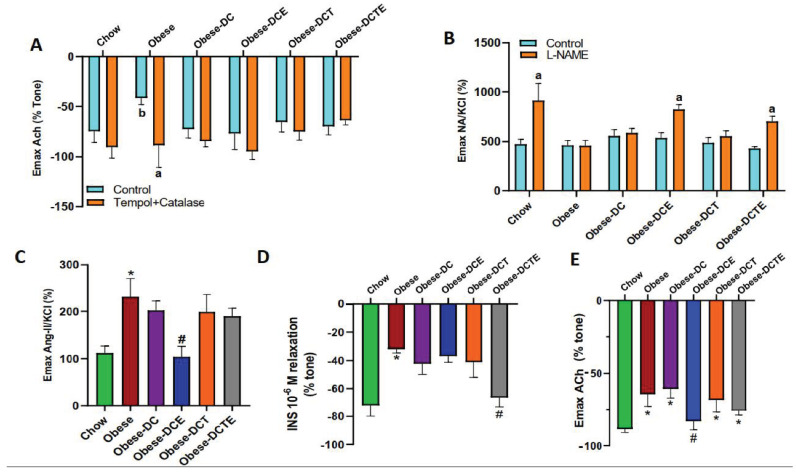
Maximum vascular response of abdominal aortic rings to accumulative concentrations of angiotensin-II (10^−11^ to 10^−6^ M) (**A**); (**B**) Dose-response curve of aorta segments to accumulative concentrations of norepinephrine (10^−9^ to 10^−4^ M) in the presence/absence of the inhibitor L-NAME (**B**); (**C**) Maximum vascular response of thoracic aortic rings to accumulative concentrations of acetylcholine (10^−9^ to 10^−4^ M) in the presence/absence of Tempol+Catalase; (**D**) Vascular response of aorta segments to a single dose of insulin (10^−6^). (**E**) Vascular response of superior mesenteric artery to accumulative concentrations of acetylcholine (10^−9^ to 10^−4^ M). Chow = mice fed a standard diet for 25 weeks; Obese = mice fed a high fat (60% kcal from fat)/high sucrose diet for 25 weeks; DC = mice fed a high fat (60% kcal from fat)/high sucrose diet for 23 weeks and switched to a diet with lower caloric intake (25% kcal from fat) for two weeks; DCE = mice fed a high fat (60% kcal from fat)/high sucrose diet for 23 weeks and switched to a diet with lower caloric intake (25% kcal from fat) supplemented with CSAT+^®^ for two weeks; DCT = mice fed a high fat (60% kcal from fat)/high sucrose diet for 23 weeks and switched to a diet with lower caloric intake (25% kcal from fat) and subjected to aerobic training for two weeks; DCTE = mice fed a high fat (60% kcal from fat)/high sucrose diet for 23 weeks and switched to a diet with lower caloric intake (25% kcal from fat) supplemented with CSAT+^®^ and subjected to aerobic training for two weeks. Values are represented as mean ± SEM; *n* = 8–10 mice/group. * *p <* 0.05 vs. chow; # *p <* 0.05 vs. obese. a *p <* 0.05 vs. their respective control; b *p <* 0.01 vs. control chow.

**Figure 9 antioxidants-11-01803-f009:**
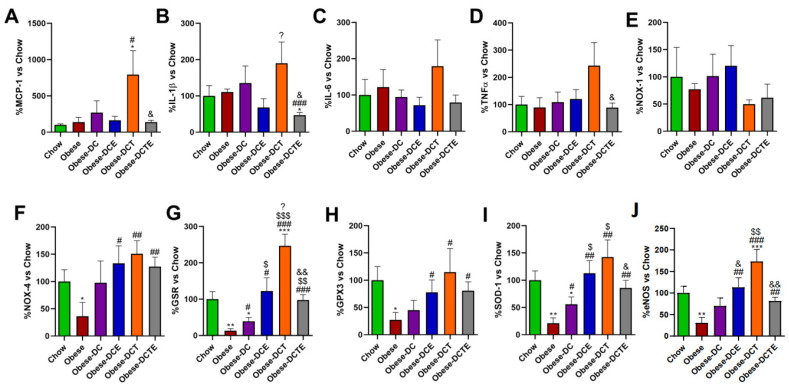
Gene expression of Monocyte Chemotactic Protein-1 (**A**), Interleukin 1β (**B**), Interleukin 6 (**C**), Tumor Necrosis Factor α (**D**) NADPH oxidase 1 (**E**), NADPH oxidase 4 (**F**), Glutathione Reductase (**G**), Glutathione Peroxidase 3 (**H**), Super Oxide Dismutase 1 (**I**) and endothelial nitric oxide synthase (eNOS) (**J**) in the aorta. Chow = mice fed a standard diet for 25 weeks; Obese = mice fed a high fat (60% kcal from fat)/high sucrose diet for 25 weeks; DC = mice fed a high fat (60% kcal from fat)/high sucrose diet for 23 weeks and switched to a diet with lower caloric intake (25% kcal from fat) for two weeks; DCE = mice fed a high fat (60% kcal from fat)/high sucrose diet for 23 weeks and switched to a diet with lower caloric intake (25% kcal from fat) supplemented with CSAT+^®^ for two weeks; DCT = mice fed a high fat (60% kcal from fat)/high sucrose diet for 23 weeks and switched to a diet with lower caloric intake (25% kcal from fat) and subjected to aerobic training for two weeks; DCTE = mice fed a high fat (60% kcal from fat)/high sucrose diet for 23 weeks and switched to a diet with lower caloric intake (25% kcal from fat) supplemented with CSAT+^®^ and subjected to aerobic training for two weeks. Values are represented as mean ± SEM; *n* = 8–10 mice/group. * *p <* 0.vs chow; ** *p <* 0.01 vs. chow; *** *p <* 0.001 vs. chow; # *p <* 0.05 vs. obese; ## *p <* 0.01 vs. obese; ### *p <* 0.001 vs. obese; $ *p <* 0.05 vs. obese-DC; $$ *p <* 0.01 vs. obese-DC; $$$ *p <* 0.001 vs. obese-DC; ? *p <* 0.05 vs. obese-DCE; & *p <* 0.05 vs. Obese-DCT; && *p <* 0.01 vs. Obese-DCT.

**Figure 10 antioxidants-11-01803-f010:**
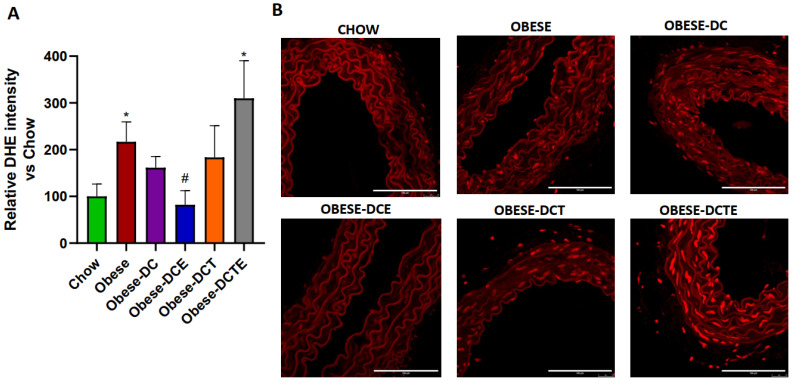
Quantification of reactive oxygen species (**A**) and representative images of aorta sections stained with dihydroethidium (**B**). The scale bar is equivalent to 100 μm. Chow = mice fed a standard diet for 25 weeks; Obese = mice fed a high fat (60% kcal from fat)/high sucrose diet for 25 weeks; DC = mice fed a high fat (60% kcal from fat)/high sucrose diet for 23 weeks and switched to a diet with lower caloric intake (25% kcal from fat) for two weeks; DCE = mice fed a high fat (60% kcal from fat)/high sucrose diet for 23 weeks and switched to a diet with lower caloric intake (25% kcal from fat) supplemented with CSAT+^®^ for two weeks; DCT = mice fed a high fat (60% kcal from fat)/high sucrose diet for 23 weeks and switched to a diet with lower caloric intake (25% kcal from fat) and subjected to aerobic training for two weeks; DCTE = mice fed a high fat (60% kcal from fat)/high sucrose diet for 23 weeks and switched to a diet with lower caloric intake (25% kcal from fat) supplemented with CSAT+^®^ and subjected to aerobic training for two weeks. Values are represented as mean ± SEM; *n* = 8–10 mice/group. * *p <* 0.05 vs. chow; # *p <* 0.05 vs. obese.

**Table 1 antioxidants-11-01803-t001:** Weights of heart, epididymal adipose tissue, lumbar subcutaneous adipose tissue, interescapular brown adipose tissue, retroperitoneal visceral adipose tissue, perivascular adipose tissue, kidneys, adrenal glands, spleen, liver, gastrocnemius and soleus muscles. Chow = mice fed a standard diet for 25 weeks; Obese = mice fed a high fat (60% kcal from fat)/high sucrose diet for 25 weeks; DC = mice fed a high fat (60% kcal from fat)/high sucrose diet for 23 weeks and switched to a diet with lower caloric intake (25% kcal from fat) for two weeks; DCE = mice fed a high fat (60% kcal from fat)/high sucrose diet for 23 weeks and switched to a diet with lower caloric intake (25% kcal from fat) supplemented with CSAT+^®^ for two weeks; DCT = mice fed a high fat (60% kcal from fat)/high sucrose diet for 23 weeks and switched to a diet with lower caloric intake (25% kcal from fat) and subjected to aerobic training for two weeks; DCTE = mice fed a high fat (60% kcal from fat)/high sucrose diet for 23 weeks and switched to a diet with lower caloric intake (25% kcal from fat) supplemented with CSAT+^®^ and subjected to aerobic training for two weeks. Values are represented as mean ± SEM; *n* = 8–10 mice/group. * *p <* 0.05 vs. chow; ** *p <* 0.01 vs. chow; *** *p <* 0.001 vs. chow; # *p <* 0.05 vs. obese; ## *p <* 0.01 vs. HFHS; ### *p <* 0.001 vs. HFHS; $ *p <* 0.05 vs. obese-DC; $$ *p <* 0.05 vs. obese-DC; $$$ *p <* 0.001 vs. obese-DC.

Weight (mg/cm)	Chow	Obese	Obese-DC	Obese-DCE	Obese-DCT	Obese-DCTE
Heart	100.5 ± 3.4	110.9 ± 5.3	125.3 ± 9.3	115.0 ± 5.8	98.4 ± 12.8	106.3 ± 5.5
Epididymal adipose tissue	396.2 ± 31.1	1186.0 ± 86.6 ***	897.9 ± 46.3 ***	733.5 ± 65.3 * ###	913.2 ± 106.1 ***	907.9 ± 63.9 ***
Lumbar subcutaneous adipose tissue	159.1 ± 28.1	1343.4 ± 45.32 ***	1020.0 ± 71.6 *** #	661.0 ± 99.7 *** ### $	799.8 ± 97.5 *** ### $	826.2 ± 81.8 *** ### $
Interescapular brown adipose tissue	55.4 ± 4.3	211.0 ± 21.8 ***	156.8 ± 19.1 *** #	124.2 ± 14.3 * ##	126.9 ± 12.8 * ## $	120.6 ± 9.2 * ### $
Retroperitoneal visceral adipose tissue	244.2 ± 19.7	1089.0 ± 71.7 ***	717.8 ± 43.2 *** ###	556.9 ± 71.6 ** ### $	551.9 ± 50.9 ** ### $	594.6 ± 35.4 *** ### $
Perivascular adipose tissue (PVAT)	7.44 ± 0.48	23.90 ± 2.31 ***	16.32 ± 2.38 *	12.46 ± 1.74 ##	13.6 ± 2.22 ##	13.8 ± 1.15 ##
Kidneys	157.4 ± 1.6	157.4 ± 1.6	191.3 ± 6.6 **	172.8 ± 7.2	175.1 ± 5.7	166.9 ± 7.6 #
Adrenal glands	1.17 ± 0.12	2.36 ± 0.21 ***	1.70 ± 0.11 ##	1.55 ± 0.12 ###	1.64 ± 0.11 ##	1.95 ± 0.09 ***
Spleen	33.8 ± 1.64	51.5 ± 3.18 ***	50.7 ± 2.09 ***	41.8 ± 1.21 # $	41.1 ± 2.4 # $	37.8 ± 1.32 ### $$$
Liver	595.7 ± 11.4	1418.1 ± 127.3 ***	948.7 ± 54.2 ** ###	748.0 ± 41.6 ### $	816.6 ± 72.2 ###	691.2 ± 25.7 ### $
Soleus	5.30 ± 0.21	6.77 ± 0.51 **	6.67 ± 0.40 **	6.10 ± 0.49	6.54 ± 0.26 ***	5.65 ± 0.31 # $
Gastrocnemius	68.6 ± 1.82	79.6 ± 4.24 **	75.7 ± 1.48 *	79.5 ± 3.84 *	73.6 ± 1.26 *	69.5 ± 1.49 # $$

**Table 2 antioxidants-11-01803-t002:** Plasma levels of total lipids, triglycerides, total cholesterol, low-density lipoproteins (LDL)-cholesterol, high-density lipoproteins (HDL)-cholesterol, glucose, leptin, insulin, adiponectin, and interleukin-6 (IL-6) and mean blood pressure. Chow = mice fed a standard diet for 25 weeks; Obese = mice fed a high fat (60% kcal from fat)/high sucrose diet for 25 weeks; DC = mice fed a high fat (60% kcal from fat)/high sucrose diet for 23 weeks and switched to a diet with lower caloric intake (25% kcal from fat) for two weeks; DCE = mice fed a high fat (60% kcal from fat)/high sucrose diet for 23 weeks and switched to a diet with lower caloric intake (25% kcal from fat) supplemented with CSAT+^®^ for two weeks; DCT = mice fed a high fat (60% kcal from fat)/high sucrose diet for 23 weeks and switched to a diet with lower caloric intake (25% kcal from fat) and subjected to aerobic training for two weeks; DCTE = mice fed a high fat (60% kcal from fat)/high sucrose diet for 23 weeks and switched to a diet with lower caloric intake (25% kcal from fat) supplemented with CSAT+^®^ and subjected to aerobic training for two weeks; HOMA-IR = Homeostatic Model Assessment for Insulin Resistance index. Values are represented as mean ± SEM; *n =* 8–10 mice/group. * *p <* 0.05 vs. chow; ** *p <* 0.01 vs. chow; *** *p <* 0.001 vs. chow; # *p <* 0.05 vs. obese; ## *p <* 0.01 vs. Obese; ### *p <* 0.001 vs. Obese; $ *p <* 0.05 vs. obese-DC; $$ *p <* 0.05 vs. obese-DC.

	Chow	Obese	Obese-DC	Obese-DCE	Obese-DCT	Obese-DCTE
Total lipids (mg/dL)	26.6 ± 9.5	253.1 ± 23.1 ***	91.2 ± 28.3 * ###	48.5 ± 15.2 ###	79.8 ± 15.2 ** ##	113.0 ± 33.5 * ##
Triglycerides (mg/dL)	42.1 ± 2.8	98.4 ± 8.7 ***	81.0 ± 6.9 ***	60.1 ± 4.5 ** ##	68.4 ± 5.5 *** #	77.7 ± 6.6 *** #
Total cholesterol (mg/dL)	152.4 ± 10.2	390.4 ± 21.8 ***	254.4 ± 21.0 ** ###	181.3 ± 22.1 ### $	225.6 ± 29.7 * ###	167.4 ± 5.9 * ### $$
LDL-c (mg/dL)	55.7 ± 5.2	140.5 ± 9.4 ***	86.0 ± 7.5 ** ###	55.1 ± 6.9 ### $$	72.4 ± 10.9 ###	45.6 ± 3.2 ### $$
HDL-c (mg/dL)	31.7 ± 2.1	68.0 ± 3.6 ***	36.4 ± 2.7 ###	37.6 ± 3.7 ###	42.3 ± 4.0 * ###	32.4 ± 2.6 ###
Glucose (mg/dL)	86.3 ± 10.8	125.6 ± 7.2 *	105.3 ± 5.5	79.7 ± 9.3 ##	101.8 ± 8.2	93.4 ± 7.2 #
Leptin (ng/dL)	4.1 ± 0.8	49.8 ± 1.8 ***	30.3 ± 2.4 *** ###	16.5 ± 3.6 * ### $$	23.8 ± 3.7 *** ###	20.8 ± 2.4 *** ###
Insulin	1.47 ± 0.32	14.39 ± 2.32 ***	7.06 ± 0.94 * ##	3.31 ± 0.64 ###	4.49 ± 0.70 ###	5.38 ± 0.97 ###
Adiponectin (ng/mL)	12.017 ± 684	62.856 ± 316 ***	6993 ± 327 ***	7590 ± 275 *** #	7452 ± 452 ***	8079 ± 420 *** #
IL-6 (pg/mL)	19.71 ± 2.34	41.6 ± 4.95 **	34.4 ± 6.96	26.4 ± 3.91 #	36.8 ± 10.2	24.5 ± 1.14 ###
HOMA-IR	0.49 ± 0.15	4.35 ± 0.75 ***	1.65 ± 0.88 ###	0.65 ± 0.15 ###	1.13 ± 0.20 ###	1.40 ± 0.32 ###
Adiposity Index	1590 ± 101	3970 ± 206 ***	3034 ± 149 *** ##	2581 ± 244 ** ###	2527 ± 172 ** ###	2885 ± 144 *** ###
Mean blood pressure (mmHg)	103.6 ± 2.1	125.7 ± 2.1 ***	118.9 ± 1.5 ***	111.6 ± 2.4 ### $	116.0 ± 3.2 **	119.1 ± 1.9 ***

## Data Availability

Data is contained within the article.

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
