# Peer review of "Carob Extract Supplementation Together with Caloric Restriction and Aerobic Training Accelerates the Recovery of Cardiometabolic Health in Mice with Metabolic Syndrome"

_antioxidants, 2022, doi:10.3390/antiox11091803_

Round 1
Reviewer 1 Report
I have reviewed the article entitled "Carob extract supplementation together with caloric restriction and aerobic training accelerates the recovery of cardiometabolic 3 health in mice, with metabolic syndrome". In my opinion this manuscript is suitable for publication in the Antioxidants Journal with corrections:
-In the scientific name of the species, Ceratonia siliqua, the initial of the botanist who described it (L.) should not be written in italics. Please consider modifying it.
-You must report all citations present in material and methods, especially for items 2.2 - 2.9
-Lines 614-615: "the presence of antioxidant compounds in 614 CSAT+® [18] may also be involved in its insulin sensitizing effects in peripheral tissues." Please specify what antioxidant compounds.
-Line 648: "The results obtained after the preincubation of aorta rings with an antioxidant". You must report.
-Lines 665-666: "Both interventions, CSAT® supply- 665 mentation and aerobic training, exert synergistic effects in certain parameters". Please specify this fact.
-The authors should refer to the main chemical compounds found in carob extract or carob tree seeds and try to discuss the activity in relation to the reported compounds.
Author Response
I have reviewed the article entitled "Carob extract supplementation together with caloric restriction and aerobic training accelerates the recovery of cardiometabolic 3 health in mice, with metabolic syndrome". In my opinion this manuscript is suitable for publication in the Antioxidants Journal with corrections:
-In the scientific name of the species, Ceratonia siliqua, the initial of the botanist who described it (L.) should not be written in italics. Please consider modifying it.
It has been corrected as suggested by the reviewer.
-You must report all citations present in material and methods, especially for items 2.2 - 2.9
As suggested by the reviewer, we have now included citations of all the methods described in sections 2.2-2.9 (references 18-27).
-Lines 614-615: "the presence of antioxidant compounds in CSAT+® [18] may also be involved in its insulin sensitizing effects in peripheral tissues." Please specify what antioxidant compounds.
As suggested by the reviewer, information regarding the antioxidant compounds present in CSAT+® has been added (Line 631).
“Moreover, since oxidative stress also plays a pivotal role in the development of insulin resistance [57], the presence of antioxidant compounds in CSAT+®, such as gallotannins and flavonols [18], may also be involved in its insulin sensitizing effects in peripheral tissues.”
-Line 648: "The results obtained after the preincubation of aorta rings with an antioxidant". You must report.
As suggested by the reviewer, the antioxidants used in the vascular reactivity experiments (tempol and catalase) are now also mentioned in the Discussion section (Line 668).
“The results obtained after the preincubation of aorta rings with the antioxidants tempol and catalase demonstrates that the beneficial effects of both exercise and CSAT+® supplementation on endothelial function are due to their antioxidant effects.”
-Lines 665-666: "Both interventions, CSAT® supplemmentation and aerobic training, exert synergistic effects in certain parameters". Please specify this fact.
As suggested by the reviewer, the synergistic effects among CSAT+® supplementation and aerobic training are now also mentioned in the conclusion section (lines 686-688):
“ Both interventions, CSAT+® supplementation and aerobic training, exert synergistic effects in preventing the obesity-induced increase in kidney weight, vascular insulin resistance, and changes in some pro-inflammatory and oxidative genes in the liver and the gastrocnemius.”
-The authors should refer to the main chemical compounds found in carob extract or carob tree seeds and try to discuss the activity in relation to the reported compounds.
We appreciate the reviewer's suggestion. References to studies conducted with previously identified compounds in CSAT+® have been added throughout the discussion section:
Lines 602-604: Similarly, it is reported that some of the compounds identified in CSAT+®, such as quercetin and gallic acid [18], also exert beneficial effects on the lipid profile in over-weight subjects [45] and in obese rodents [46,47].
Lines 613-615: Based on previous studies, this anti-inflammatory effect may be conferred, at least in part, by certain compounds present in CSAT+® such some flavonols and gallic acid [46,50].
Lines 632-633: Indeed, gallic acid and quercetin are reported to improve glucose tolerance both in humans and animal models of diet-induced obesity [46,62,63].”
Lines 662-663: This effect may be produced, at least in part, by quercetin which exerts vasodilating effects through the increase in NO availability [66,67].

Reviewer 2 Report
This study analyzed whether supplementation with carob fruit extract (CSAT+®), alone or in combination with aerobic training, accelerates the recovery of cardiometabolic health in mice with MetS subjected to caloric restriction. Overall, the study was well executed, and the results were appropriately interpreted.
I have only made a few comments.
1. What are the energy supply ratios of macronutrients for the standard chow, HFHS diet, and low-calorie diet (for treatment)?
2. Has it been determined that the mice obesity or MetS model has been successfully established? What are the criteria for success? Please clarify in the manuscript.
3. Please supplement the calculation of the sample size in the methods section.
4. Table 1. “Weight(mg/cm)” which indicates the weight of the viscera and adipose tissue per cm?
5. Line 359-360: The description is not completely consistent with Figure 3.
There are also minor issues. Some illustrative examples:
Line 86: “in mice with.”
Line 142: “12.500 rpm”
Lines 177 and 471: “PGC-1α”, “L-NAME” has no corresponding full name when it first appears in the manuscript.
Line 228: “processedin”
Line 317: “form”
Line 332: “## p<0.01 vs HFHS”
Line 355: “* p<0,05 vs insulin; ** p<0,01 vs insulin”, orange represents the insulin presence group, why is it still compared with insulin?
Commas were used for decimal points in some tables.
Author Response
This study analyzed whether supplementation with carob fruit extract (CSAT+®), alone or in combination with aerobic training, accelerates the recovery of cardiometabolic health in mice with MetS subjected to caloric restriction. Overall, the study was well executed, and the results were appropriately interpreted.
I have only made a few comments.
- What are the energy supply ratios of macronutrients for the standard chow, HFHS diet, and low-calorie diet (for treatment)?
The energy supply ratios of macronutrients for the different diets are the following:
|
|
Standard chow |
HFHS |
Diet with 25% kcal from fat |
Diet with 25% kcal from fat + CSAT (4.8%) |
|
Protein (g%) |
24 |
23 |
17,3 |
17,2 |
|
Carbohydrates (g%) |
65 |
35,5 |
55,1 |
54,9 |
|
Fats (g%) |
11 |
35,8 |
10,8 |
10,7 |
- Has it been determined that the mice obesity or MetS model has been successfully established? What are the criteria for success? Please clarify in the manuscript.
As it is shown in Tables 1 and 2, we are quite convinced that our obese mice suffered from MetS since they showed several alterations characteristic of subjects with MetS such as:
- Increased visceral adiposity as shown by the increased weights of epidydimal and retroperitoneal adipose tissues
- Hypertriglyceridemia (98,4±8,7 mg/dl)
- Hyperglycaemia under fasting conditions (125,6±7,2 mg/dl)
- Hypertension
- Increased circulating levels of LDL-c (140,5±9,4 mg/dl)
To clarify this issue the following paragraph has been added to the manuscript (Lines 132-133)
“MetS was confirmed by the presence of visceral adiposity, hypertriglyceridemia, hyperglycaemia, hypertension and increased circulating levels of LDL-c.”
- Please supplement the calculation of the sample size in the methods section.
The sample size calculation was carried out based on previous studies from our group in which the same parameters were analyzed. The G*Power program was used to perform an a priori analysis of a one-way ANOVA with 6 experimental groups. The value assumed for the size effect was 0.65, for the significance (a) was 0.5 and for the power (b), 0.95.
This paragraph has been added to the material and methods section (Lines 112-116).
- Table 1. “Weight(mg/cm)” which indicates the weight of the viscera and adipose tissue per cm?
Weight (mg/cm) indicates the weight of the organ (in mg) compared to the length of the tibia (in cm). In this way, we correct the weight of the organ according to the size of the animal. This representation is more accurate to measure the weights or organ and tissues than the relative body weight (% of body weight) which is not recommended in models in which the body weight of the animals is highly affected.
To clarify this issue the following sentence has been added (line 271):
“Data of organ weights respect to the length of the left tibia of mice from the different experimental groups are represented in Table 1.”
- Line 359-360: The description is not completely consistent with Figure 3.
We thank the reviewer for having detected this error. It has been corrected.
There are also minor issues. Some illustrative examples:
Line 86: “in mice with.”
Line 142: “12.500 rpm”
Lines 177 and 471: “PGC-1α”, “L-NAME” has no corresponding full name when it first appears in the manuscript.
Line 228: “processedin”
Line 317: “form”
Line 332: “## p<0.01 vs HFHS”
Line 355: “* p<0,05 vs insulin; ** p<0,01 vs insulin”, orange represents the insulin presence group, why is it still compared with insulin?
Commas were used for decimal points in some tables.
We thank the reviewer for the indications. All the mentioned errors and others found throughout the article have been corrected and marked in red.
